# Inappropriateness of Repeated Laboratory and Radiological Tests for Transferred Emergency Department Patients

**DOI:** 10.3390/jcm8091342

**Published:** 2019-08-29

**Authors:** Jérôme Bertrand, Christophe Fehlmann, Olivier Grosgurin, François Sarasin, Omar Kherad

**Affiliations:** 1Emergency Department, Department of Acute Medicine, Geneva University Hospitals, CH-1205 Geneva, Switzerland; 2Department of General Internal Medicine, Department of Medicine, Geneva University Hospitals, CH-1205 Geneva, Switzerland; 3Department of Internal Medicine and University of Geneva, La Tour Hospital, CH-1217 Geneva, Switzerland

**Keywords:** quality, redundant, less-is-more, choosing wisely, emergency medicine, laboratory test, radiological procedure

## Abstract

Background: Laboratory and radiographic tests are often repeated during inter-hospital transfers from secondary to tertiary emergency departments (ED), despite available data from the sending structure. The aim of this study was to identify the proportion of repeated tests in patients transferred to a tertiary care ED, and to estimate their inappropriateness and their costs. Methods: A retrospective chart review of all adult patients transferred from one secondary care ED to a tertiary care ED during the year 2016 was carried out. The primary outcome was the redundancy (proportion of procedure repeated in the 8 h following the transfer, despite the availability of the previous results). Factors predicting the repetition of procedures were identified through a logistic regression analysis. Two authors independently assessed inappropriateness. Results: In 2016, 432 patients were transferred from the secondary to the tertiary ED, and 251 procedures were repeated: 179 patients (77.2%) had a repeated laboratory test, 34 (14.7%) a repeated radiological procedure and 19 (8.2%) both. Repeated procedures were judged as inappropriate for 197 (99.5%) laboratory tests and for 39 (73.6%) radiological procedures. Conclusion: Over half of the patients transferred from another emergency department had a repeated procedure. In most cases, these repeated procedures were considered inappropriate.

## 1. Introduction

Health expenditure is rising in most OECD (Organisation for Economic Co-operation and Development) countries, both in absolute terms and as a proportion of the gross domestic product (GDP) (12% in Switzerland in 2015) [1]. However, it is estimated that 20% of medical procedures are at best ineffective and at worst wasteful and harmful. Compelling evidence, summarized in Choosing Wisely recommendations, demonstrates that this waste can be minimized in our healthcare systems [2,3]. Duplicate services, such as the inappropriate redundancy of procedures during patients’ transfer from one emergency department (ED) to another, were addressed in a recent OECD report that tackled spending in healthcare systems by defining different types of wasteful medical activities [4,5].

Depending on the access to specialized medical resources, primary and secondary hospitals often transfer patients to tertiary care ED. These inter-hospital transfers represent a critical period, exposing patients to the potentially negative consequences of handoffs on the consumption of medical resources [6,7].

There is only limited medical literature on the frequency and the appropriateness of repeated tests following inter-hospital transfers [8,9]. However, emergency structures, which mostly represent the gateway to healthcare institutions, could benefit from a review of these practices both in terms of economic efficiency and quality of care.

The purpose of this study was to estimate the frequency of redundant tests, the proportion of inappropriateness, and the direct costs of repeated procedures when patients were transferred from a secondary hospital to a tertiary hospital.

## 2. Materials and Methods

### 2.1. Design and Setting

We conducted a retrospective chart review of all patients transferred from one secondary ED (Hôpital La Tour (HLT)) to a tertiary ED (the Geneva University Hospitals (HUG)) from January 2016 to December 2016, in Geneva (Switzerland). This study was carried out in two of the main EDs of the canton of Geneva, where emergencies are currently organised in a network (Geneva Emergency Network), including several clinics and hospitals in the canton of Geneva. Each centre is likely to receive patients according to its competences and its access to specialized care. HUG is a tertiary and reference structure and the only trauma centre in the canton (72,921 visits in 2018). HLT is a secondary academically affiliated hospital and the second largest ED in the canton (23,343 visits in 2018). However, HLT cannot currently guarantee specialized consultations for all disciplines and must, as a private structure, deal with insurance-related constraints limiting the access of admission. As a result, a number of patients are referred to the HUG each year through the ED. These two hospital structures have an electronic health record (EHR) and share a common computer platform for imaging data (PACS, Picture Archiving and Communication System). On the other hand, HLT EHR is not accessible by the providers of HUG and vice versa, without the possibility to import automatically data from one EHR to the other. In clinical practice, the referring hospital prints a medical report with all the information (history, vitals and laboratory data) that is given either directly to the patient or to the paramedics who transferred the patient, ensuring that the medical data is almost always available. A triage nurse or an administrative clerk at the receiving hospital calls the referring hospital to fax them the medical reports if they were not given to the patient.

### 2.2. Population

This study included all patients over 18 years old transferred from HLT to the HUG from January to December 2016. Exclusion criteria where patients transferred to gynaecology-obstetrics, Ear Nose and Throat (ENT), psychiatry or ophthalmology ED, as well as patients who were directly transferred to an inpatient unit or to the intensive care, without accessing through the ED. Patients who never came in the second ED, patient with missing HER in one of the two centres and duplicate entry were also excluded.

### 2.3. Data Collection

We used an administrative database referencing all transfers from HLT to the HUG ED over the studied period. For the purpose of our study, the principal investigator (J.B.) had access to both EHR. Transferred patients were identified using a list from the referring hospital, extracted from their EHR using the code “transferred to Geneva University Hospital”. This list included administrative information (name, date of birth) and was matched with EHR from the receiving hospital. JB performed a chart review at the receiving hospital and verified that the medical report from the referring hospital was available. During this chart review, J.B. documented the date of transfer, the reason for transfer, the time of transfer (day, night, working days or not), the main final diagnosis (by organ category), the length of stay at the ED, the type of pathology and associated care (medical versus surgical), the presence or absence of repeated procedures and the reason of the repetition if available. Repeated procedures could be laboratory tests and/or radiological procedures. Laboratory tests included simple (haemoglobin, white blood cells, platelets) or full blood counts, coagulation (TP, PTT, DDimers), chemistry (electrolytes, renal function, liver tests, lipase, C-reactive protein, troponins). Urinary analyses and other more specific tests (blood cultures, urinary antigen, thick smear, etc.) were also included. Radiological procedures included conventional radiology (lung, bone), ultrasound and CT-Scan.

### 2.4. Outcomes Measure

The primary outcome was the redundancy, namely the proportion of repeated procedures, defined as a procedure repeated during the 8 h following the transfer despite the availability of the previous results.

The secondary outcomes were the inappropriateness and the direct costs of the repeated procedures. The appropriateness was assessed independently by two authors (O.G. and O.K.), based on their clinical experience and on the clinical context (reason for consultation, reason for transfer, diagnosis made at HLT, diagnosis made at the HUG, length of stay, reason for repetition, etc.). A repeated laboratory test was considered as inappropriate if, regardless of the initial result, it did not influence the patient’s therapeutic management or referral (for example, in a digestive bleeding, repetition of haemoglobin was appropriate, but repetition of creatinine was not). A repeated radiological procedure was considered as inappropriate if repeated identically in both centres (i.e., without additional information such as another incidence or injection of contrast material) without change in the clinical condition. The term “partially” inappropriate applied when part of the repetition of the laboratory tests seemed justified, while the other part was not.

The measurement of the direct cost of these inappropriate procedures only included the direct cost of the test. This cost was calculated for radiology exams on the basis of the rates applied in the HUG, after confirmation with the invoicing department. For laboratory tests, each item or group of laboratory items was associated with a fixed value (e.g., 10 CHF for the CRP assay). The total cost was therefore obtained by adding the set of values associated with the repeated tests. This cost, therefore, did not take into account indirect costs, such as nursing services, time required to complete exams, equipment costs, etc.

### 2.5. Statistical Analysis

The student *t*-test or Wilcoxon rank-sum test and the Chi-squared test for group comparisons were used when appropriate. The inter-observer agreement was measured with a Kappa-coefficient. A 2-sided *p*-value below 0.05 was considered significant. For multivariable model, variables were chosen based on clinical relevance and included independently of univariable regression’s results, with a ratio of 10 events per variable ratio. Statistical analysis was performed using STATA version 15 (Stata Corporation, Texas, TX, USA).

This study was approved by the Cantonal Commission on Research Ethics (CCER) of Geneva, Switzerland.

## 3. Results

In 2016, 549 patients were transferred from HLT to the HUG. Among them, 117 files were excluded: Transfers to other speciality ED (ENT (12), gynaecology/obstetrics (26), ophthalmology (8), paediatrics (33) and psychiatry (10)), direct hospitalisations (3), duplicate entries (2) and missing EHR (23). Eventually, 432 patients were included.

Patients were 55.5 ± 21.1 years old and 58.6% were males (Table 1). Transfers were more frequent during the day (56.3%) and during weekends (16.7% on a Saturday or a Sunday versus 13.3% on a business day). Overall, 190 patients (44.0%) had a medical pathology and 234 (54.2%) had a surgical pathology (information missing for eight patients (1.9%)).

For the primary outcome, 232 (53.7%) patients had a repeated procedure within 8 h of the transfer: 179 (77.2%) had a repeated laboratory test, 34 (14.7%) a repeated radiological procedure and 19 (8.2%) had both. Only patients with neurology or various pathology had a redundant exam in less than half of the cases. After adjustment for gender, age, typology and time of transfer (night, weekend), the independent factor for having a repeated laboratory test was age greater than 65 years (OR = 1.8, 95% CI 1.2–2.7, *p* = 0.007) (Table 2). The odds of having a repeated radiological procedure were almost two-fold higher in patients with surgical pathology (OR = 1.9, 95% CI 1.0–3.7, *p* = 0.040) (Table 2).

For the secondary outcome, 197/198 (99.5%) of the laboratory tests and 39/53 (73.6%) of the radiological procedures were considered partially or totally inappropriate. The two reviewers had an agreement of 88.9% for laboratory tests (disaccord on 22 tests, Kappa = 0.57) and of 92.5% for radiological procedures (disaccord on four procedures, Kappa = 0.82). Table 3 summarises the main final diagnosis categories and repeated tests for patients with totally or partially inappropriate redundant tests. Among the 12 patients with confirmed pneumonia, eight had inappropriate redundant tests (for example, four chest *x*-rays, seven CRP) and among the 59 patients with confirmed bone fracture, 19 had inappropriate redundant tests (for example, 16 new *x*-ray without reduction or immobilisation, four platelets counts). The direct cost caused by redundant tests was calculated at CHF 20,000 for the year (11,900 for laboratory tests and 8100 for radiological procedures), without including ancillary costs (nursing services, equipment, etc.).

## 4. Discussion

This study reveals that over half of the patients (53.7%) transferred from another emergency department had a repeated procedure within the 8 h following transfer. In most cases (94%), these repeated procedures were considered as partially or totally inappropriate.

These data mirror an American study published by Rogg et al. that described a proportion of redundant tests (blood laboratory and urinalysis) after inter-hospital transfer in 46% and 100% of cases, respectively [7]. Noteworthily, this study did not assess the appropriateness of the redundant test by integrating the clinical circumstances that led to ordering a service: If the repeated test within 8 h was normal, it was considered as potentially inappropriate. Thus, to the best of our knowledge, our study is the first to attempt to integrate the clinical context through a critical review of the medical charts. The high concordance rate of observer highlights the inappropriateness of the procedures, which represents a real wasteful medical activity according the OECD definition [4].

Repeating laboratory tests inappropriately is not only wasteful but also painful, expensive and can also have consequences that have been described in previous studies, such a hospital-acquired anaemia, which has been associated, in moderate and severe cases, with higher long-term mortality and greater diseases severity (in particular in cases of acute myocardial infarction and delayed recovery) [10,11,12]. Furthermore, the repetition of radiological procedures may expose the patient to potential risks, associated with low dose exposure to ionising radiation, as highlighted by large epidemiological studies [13]. This can be prevented by education of providers about the impact of unnecessary prescriptions on patients and the need to emphasize the legitimacy of these procedures in terms of clinical orientation and care [12]. Sharing evidence around inappropriate medical care is critical in raising physician awareness and encouraging behaviour change.

This study was not designed to understand the reasons why the ED providers repeat these procedures inappropriately after an inter-hospital transfer. We can hypothesize that the access to information during transfers may be a real issue in a fragmented healthcare system such as the Swiss system [14]. Prior work suggested that inter-hospital transfers may represent a critical period with the risk of discontinuation of care which may lead to delays or errors [15]. The transfer of a patient to ED per se, is also an identified risk factor for the discontinuity of care and the loss of information, with potential consequences on the quality of care, the proportion of repeated exams and the length of stay [16]. ED providers may face barriers to optimal handoffs, including a lack of shared communication standards and difficulty accessing external medical records. Subsequently, handoffs require standardised transfer procedures (clinical documentation, communication between caregivers, accessibility of data, etc.) in order to avoid risks of discontinuity of care on clinical outcomes [5,6].

However, the existing EHR at HLT allows a rapid documentation of the performed test. Medical reports, including laboratory tests, are brought during transfers, limiting the readability issue of hand-written reports. Even if we cannot ensure that the medical report was available during the transfer due to the retrospective design of this study, which is a major limitation, we are confident that the data were at the disposal of the attending physician. Indeed, in clinical practice, the referring hospital prints a medical report with all the information (history, vitals, and laboratory data) that is given either directly to the patient or to the paramedics who transferred the patient (it is mandatory for paramedics to get the discharge summary and lab results before caring for the patient). Furthermore, a nurse or an administrative clerk must call the referring hospital to fax them the report if they were not given to the patient. As a last resort, residents can call to receive the results. In addition, PACS between the HUG and HLT enables the sharing of all radiological procedures carried out in the sending centre. Therefore, we can even hypothesise that the rate of inappropriately repeated tests reported in our study is underestimated within the Geneva Emergency Network; this network of emergency structures in Geneva includes several ED structures that do not have easily readable EHR and information on laboratory tests and radiological procedures performed are even less accessible.

The habits ED providers develop during their training may also drive up the inappropriate redundancy of these procedures. In their collective unconscious, there is a belief that the results would be less reliable or at least different from one centre to another. Other studies have shown a tendency to repeat biological tests during inter-hospital transfers, partly due to the so-called “group” prescription of entire panels of tests, leading to involuntary redundancies [7]. In our study, this explanation cannot be retained since the computerized prescription can be completely dissociated individually, item by item.

The costs generated by these repeated procedures were estimated on the basis of the unit rates charged at the HUG at CHF 20,000 over one year. The laboratory test only concerns the actual invoice, and therefore underestimates the real cost, including nursing services, the use of sterile equipment, catheter placements, etc. In addition, this study only took into account transfers from HLT, which, as mentioned above, has a common interface (PACS) with the HUG, as well as an easily readable EHR. If we had included transfers from all the other structures in the RUG with less accessible documentation systems, the rate of redundancy could be higher, leading to higher costs.

In this study, the medical procedure was considered as redundant if it was repeated within 8 h after the transfer. This 8-h delay was arbitrarily defined by the investigators, taking into account their clinical experience and was based on pre-existing studies [8]. However, by definition, the clinical condition of a patient in the emergency room is likely to vary rapidly. Regarding the clinical context, this 8-h delay could be considered too long and may artificially increase the window within which a test is considered redundant. Contrasting with previous data, our study is the first to integrate the clinical context in order to assess whether a test was appropriate or not, limiting this issue.

However, appropriateness was based on the review of selected elements of the medical file (reason for transfer, diagnoses retained in the various centres, tests carried out in the two centres) and not on the whole file, with a risk of interpretation bias. It did not seem to represent a major issue, given the high correlation rate between evaluators. In order to increase the generalisability of the results and to explore the reasons behind inappropriate repetitions, a qualitative prospective study, including several ED, integrating the clinical context and examining the physician ordering practices, is urgently warranted.

## 5. Conclusions

This local study identified wasteful medical activities during transfers in ED. The inappropriate repetition of laboratory and/or radiological tests not only led to an increase in costs, with an obvious impact in terms of economy, but may also have had potential negative consequences in terms of quality of care and unnecessary ionising radiations. This theme is perfectly in line with the concept of “less is more” promoted by the Choosing Wisely campaign, which focuses on the value of care and potential risks to patients, rather than using cost as a motivating factor.

## Figures and Tables

**Table 1 jcm-08-01342-t001:** Patients’ characteristics ^1^.

Variable	Total (*n* = 432)	Patients with Redundant Tests (*n* = 232)	Patients without Redundant Tests (*n* = 200)	*p*-Value
Gender (female)—no (%)	179 (41.4)	94 (40.5)	85 (42.5)	0.651
Age—year (mean, SD)	55.4 ± 21.1	58.3 ± 20.9	52.1 ± 21.0	0.002
Age—no (%)				
>65	158 (36.6)	96 (41.4)	62 (31.0)	0.026
Patient typology—no (%) ^2^				0.675
Surgical	234 (54.2)	129 (55.6)	105 (52.5)	
Medical	190 (44.0)	98 (42.2)	92 (46.0)	
Missing values	8 (1.9)	5 (2.2)	3 (1.5)	
Time of transfer—no (%)				
Night	189 (43.8)	101 (43.5)	88 (44.0)	0.923
Weekend	144 (33.3)	75 (32.3)	69 (34.5)	0.633
Main final diagnosis (by organ category)—no (%)				<0.001
Digestive pathology *Appendicitis* *Digestive bleeding* *Pancreatitis* *Cholecystitis*	99 (22.9) 31 (7.2) 12 (2.8) 15 (3.5) 11 (2.6)	77 (33.2) 23 (9.9) 12 (5.2) 10 (4.3) 9 (3.9)	22 (11.0) 8 (4.0) 0 (0.0) 5 (2.5) 2 (1.0)	
Orthopaedic pathology *Fracture*	74 (17.1) 59 (13.7)	38 (16.4) 27 (11.6)	36 (18.0) 32 (16.0)	
Neurologic pathology *TIA/Stroke*	89 (20.6) 32 (7.4)	26 (11.2) 12 (5.2)	63 (31.5) 20 (10.0)	
Pulmonary problem *Pneumonia*	26 (6.0) 12 (2.8)	21 (9.1) 8 (3.5)	5 (2.5) 4 (2.0)	
Urologic pathology	30 (6.9)	18 (7.8)	12 (6.0)	
Cardiologic pathology	18 (4.2)	15 (6.5)	3 (1.5)	
Various pathology ^3^	96 (22.2)	37 (16.0)	59 (29.5)	

^1^ SD = standard deviation. Percentages may not total 100 due to rounding. ^2^ In our system, at the end of the triage, the triage nurse attributes a typology of suspected diagnosis and associated care (either medical or surgical. If s/he does not have an idea, s/he can leave the field empty. ^3^ Various pathology includes diagnoses such as ENT, rheumatologic, dermatologic, hematologic, psychiatric or nephrological pathologies.

**Table 2 jcm-08-01342-t002:** Independent factors for redundant tests (laboratory and radiological).

	Laboratory Tests			Radiological Procedures		
Variable	aOR *	95% CI	*p*-Value	aOR *	95% CI	*p*-Value
Women	0.9	0.6–1.4	0.717	1.0	0.6–1.9	0.930
Surgical case	0.8	0.5–1.1	0.203	1.9	1.0–3.7	0.045
Age > 65	1.7	1.1–2.6	0.009	1.3	0.7–2.5	0.377
Night	1.1	0.7–1.6	0.683	0.9	0.5–1.7	0.862
Weekend	0.8	0.5–1.2	0.357	1.4	0.7–2.5	0.332

aOR = adjusted odds ratio, CI = confidence interval, * adjusted for gender, age, typology and time of transfer (night, weekend).

**Table 3 jcm-08-01342-t003:** Patients with partially or totally inappropriate redundant exams ^1^.

Final Diagnosis Category—No (%) ^2^	Total (*n* = 221)	RBC	WBC	PLT	Coag	CRP	Creat	Na/K	Liver’s Tests	Lipase	XR	US	CT
Digestive pathology *Appendicitis* *Digestive bleeding* *Pancreatitis* *Cholecystitis*	77 (34.8) 23 (10.4) 12 (5.4) 10 (4.5) 9 (4.1)	73 (95) 22 (96) 12 (100) 9 (90) 9 (100)	72 (94) 22 (96) 12 (100) 9 (90) 9 (100)	72 (94) 22 (96) 12 (100) 9 (90) 9 (100)	22 (29) 4 (17) 7 (58) 2 (20) 1 (11)	68 (88) 21 (91) 7 (58) 10 (100) 9 (100)	72 (94) 22 (96) 10 (83) 9 (90) 9 (100)	72 (94) 22 (96) 10 (83) 9 (90) 9 (100)	37 (48) 3 (13) 5 (42) 8 (80) 9 (100)	32 (42) 1 (4) 2 (17) 9 (90) 9 (100)	0 (0) 0 (0) 0 (0) 0 (0) 0 (0)	2 (3) 0 (0) 0 (0) 0 (0) 1 (9)	1 (1) 1 (4) 0 (0) 0 (0) 0 (0)
Orthopaedic pathology *Fracture*	30 (13.6) 19 (8.6)	12 (40) 4 (21)	11 (37) 3 (16)	12 (40) 4 (21)	4 (13) 0 (0)	6 (20) 0 (0)	9 (30) 3 (16)	9 (30) 3 (16)	1 (3) 0 (0)	0 (0) 0 (0)	20 (67) 16 (84)	0 (0) 0 (0)	0 (0) 0 (0)
Neurologic pathology *TIA/Stroke*	25 (11.3) 11 (5.0)	21 (84) 8 (73)	21 (84) 7 (64)	21 (84) 7 (64)	11 (44) 5 (45)	10 (40) 3 (27)	24 (96) 11 (100)	23 (92) 11 (100)	1 (4) 1 (9)	0 (0) 0 (0)	0 (0) 0 (0)	0 (0) 0 (0)	3 (12.0) 2 (18)
Pulmonary problem *Pneumonia*	20 (9.1) 8 (3.6)	18 (90) 7 (88)	17 (85) 7 (88)	18 (90) 7 (88)	5 (25) 1 (13)	17 (85) 7 (88)	17 (85) 7 (88)	17 (85) 7 (88)	2 (10) 1 (13)	0 (0) 0 (0)	6 (30) 4 (50)	0 (0) 0 (0)	0 (0) 0 (0)
Urologic pathology	18 (8.1)	16 (89)	14 (78)	15 (83)	4 (22)	10 (56)	14 (78)	12 (67)	1 (6)	0 (0)	0 (0)	1 (6)	0 (0)
Cardiologic pathology	15 (6.8)	15 (100)	15 (100)	14 (93)	8 (53)	12 (80)	15 (100)	15 (100)	5 (33)	3 (20)	3 (20)	0 (0)	0 (0)
Various pathology	36 (16.3)	30 (83)	28 (78)	180 (82)	63 (29)	21 (58)	29 (81)	27 (75)	12 (33)	2 (6)	7 (19)	0 (0)	0 (0)

^1^ Abbreviation: RBC = red blood cell, WBC = white blood cell, PLT = platelet, Coag = coagulation parameters, CRP = C-reactive protein, Creat = Creatinine, Na/K=sodium and/or potassium, XR = conventional *x*-ray, US = ultrasonography, CT = computerized tomography. ^2^ Percentages may not total 100 due to rounding.

## Data Availability

The data that support the findings of this study have restrictions and are therefore not publicly available. Data are however available from the authors upon reasonable request.

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
