# Peer review of "Inappropriateness of Repeated Laboratory and Radiological Tests for Transferred Emergency Department Patients"

_jcm, 2019, doi:10.3390/jcm8091342_

Round 1
Reviewer 1 Report
I had the pleasure to review “Inappropriateness of repeated laboratory and radiological tests for transferred emergency department patients” by Jérôme Bertrand et al. The authors analyzed the rate and cost of inappropriate double blood testing and radiographies of 432 patients being transferred from a second level to a third level emergency department in Geneva, Switzerland. In my opinion, this manuscript discusses a very important topic. Furthermore, the authors show reasonable evidence to back up their conclusions. However, in my opinion, the manuscript needs further improvements to be acceptable for publication:
- The methodology in detail is not completely clear to me. It should clearly be stated in the Methods section how health data from the referring hospital was normally transferred to the third-level hospital. Was there the availability to import the data from the referring hospital into the electronic health records? The authors describe that the analysis was performed independently by two authors, but they did not describe inter-observer differences. How could an electronic medical health record be “incomplete” or “non-analysable”? Why was information missing on 8 additional patients?
Furthermore, I have a few minor comments:
- Please check for spelling errors, for example “transfer” instead of “transferred” on line 49, “the” missing before “frequency” (line 56), “days” instead of “day” (line 133), “were” instead of “was” (line 140).
- It is not clear to me if the authors checked if the lab results were already available and really transferred to the other hospital at the time of referral. Otherwise, this would be a potential reason for double testing.
- I would consider describing double testing not just “wasteful” but also “harmful” (especially line 47). Did the authors analyze the risk of unnecessary exposure to radiation?
- P values in Table 1 would be interesting. Why is there no line for “medical pathology” (as it is currently NOT the opposite of a “surgical pathology”)?
- Why are the lines 92-96 italic?
- Please mention in the abstract that you included “all” adult patients transferred to the third level ED.
Author Response
Thank you for having taken the time to review our article. You will find in attachment a table with your comments, our answers and our modifications. We hope it will give you a better insight of our work.
Reviewer 2 Report
The article sent for the review is written in the right way. Although its scientific background is not too high, it can have an important practical aspect.
I think that for the practical value of the article to be real, a better description of the group of patients with inappropriately repeated tests should be better described. Although, patients that have been excluded from the study have been described, but in my opinion it is not enough.
Second, interesting will be to check, which laboratory tests have been made excessively? All the mentioned? There is a significant difference between the indications for complete blood count and, for example, coagulation parameters.
The same for radiological procedures. For example, I would like to read that in the case of pneumonia, further repeated radiological examinations were not justified. Even beter, if this will be accompanied by the ethiology of the disease. In this spirit, the article should be corrected.
In general, the less research you do, the better - it's recently quite common knowledge.
If no more data will be available, it is difficult to use this article in practice.
Author Response

(The authors gave the same response as above.)

Round 2
Reviewer 1 Report
The paper has substantially improved. However, there are still a few errors (i.e., HER instead of EHR in 2 instances) and the authors should mention the risk of repeated exposure to radiation in the conclusions.
Author Response
Thank you again for all your comments that helped us to improve our manuscript.
As you mentioned, we corrected the "HER" mistake.
Concerning the risk of repeated exposure to radiation, we addressed it in adding two sentence.
The first one in the discussion :"Furthermore, the repetition of radiological procedures may expose the patient to potential risks, associated with low dose exposure to ionising radiation, as highlighted by large epidemiological studies." The second one in the conclusion : "...potential negative consequences in terms of quality of care and unnecessary ionising radiations."
Reviewer 2 Report
The article has been corrected and I think it is interesting and clinically relevant.
Author Response
Thank you again for your comments that helped us to improve our manuscript.